# Stable isotopes as a predictor for organic or conventional classification of berries and vegetables

Xia Zhu-Barker[1]*, Michael Liou[2], Diana Zapata[3], Jingyi Huang[1], William R. Horwath[3]

**1** Department of Soil and Environmental Sciences, University of Wisconsin-Madison, Madison, Wisconsin, United States of America, **2** Department of Statistics, University of Wisconsin–Madison, Madison, Wisconsin, United States of America, **3** Department of Land, Air and Water Resources, University of California-Davis, Davis, California, United States of America

* zhubarker@wisc.edu

## Abstract

Organic agriculture is expanding worldwide, driven by expectations of improving food quality and soil health. However, while organic certification by regulatory bodies such as the United States Department of Agriculture and the European Union confirms compliance with organic standards that prohibit synthetic chemical inputs, there is limited oversight to verify that organic practices, such as the use of authentic organic fertilizer sources, are consistently applied at the field level. This study investigated the elemental content of carbon (C) and nitrogen (N) and their stable isotopes ($\delta^{13}C$ and $\delta^{15}N$) in seven different crops grown under organic or conventional practices to assess their applicability as a screening tool to verify the authenticity of organic labeled produce. Holm corrected Welch t-tests and a generalized linear mixed model (GLMM) were used to assess the potential of stable isotope or crop elemental content to differentiate organic vs. conventional production systems. Total C and N content or C/N ratio was not significantly different between production systems or among geographic origins for most crops. However, the average N stable isotope ($\delta^{15}N$) content differed, with conventional crops at 1.8 ± 2.2‰ and organic at 6.0 ± 3.4‰. A mixed model incorporating elemental contents and stable isotopes identified $\delta^{15}N$ as the primary predictor in discriminating organic and conventional production systems. A $\delta^{15}N$ threshold is suggested to differentiate conventional from organic grown raspberries ($\delta^{15}N < 2.17‰$) and strawberries ($\delta^{15}N < 3.22‰$), for an estimated false negative rate of 1%. Although further evaluation is needed, our extensive dataset (n = 791) captures key predictors of agricultural production systems and holds potential as a benchmark for future organic production verification.

## Introduction

Organic agriculture has a reputation for having fewer negative consequences to the environment, and the consumption of organic produce is considered to promote human health [1,2]. However, while organic farming has advantages, it also presents unique challenges. For produce to be certified organic, such as in the U.S., it must be grown without synthetic

**Data availability statement:** The datasets used in this study and all the scripts for data analysis were deposited in the public data Repository Figshare (DOI: 10.6084/m9.figshare.24844026 or https://figshare.com/articles/software/Stable_Isotopes_as_a_Predictor_for_Organic_or_Conventional_Classification_of_Berries_and_Vegetables_Code_Supplement_/24844026).

**Funding:** The study was supported by the Office of the Vice Chancellor for Research and Graduate Education at the University of Wisconsin-Madison with funding from the Wisconsin Alumni Research Foundation and the J.G. Boswell Endowed Chair in Soil Science. The funders had no role in study design, data collection and analysis, decision to publish, or preparation of the manuscript.

**Competing interests:** The authors have declared that no competing interests exist.

**Abbreviations:** C, carbon; N, nitrogen; C/N, carbon to nitrogen ratio; $\delta^{15}N$, the N stable isotopic signature at natural abundance; $\delta^{13}C$, the C stable isotopic signature at natural abundance; GLMM, generalized linear mixed model; AICc, Akaike Information Criteria, corrected; SS, subject-specific; PA, population-averaged; AUROC, areas under the receiver operating characteristic curve; FNR, false negative rate.

fertilizer and pesticide inputs, adhere to defined standards of production, and undergo an annual certification audit [3]. The exclusion of conventional agrochemicals in organic farming often leads to higher production costs compared to conventional farming, primarily due to lower yields, increased labor and input costs, and greater crop losses from pests and diseases [4,5]. Among these challenges, organic farming systems navigate the complexities introduced by the prohibition of synthetic inputs, which adds layers of management difficulty and economical constraints that are not typically encountered in conventional production systems [6,7]. This prohibition poses a particular challenge in aligning nitrogen (N) availability from organic sources with the timing of crop demand. The difficulty in synchronizing N availability with crop demand in organic farming systems has led to one of the most commonly reported organic violations: the use of synthetic N fertilizers [8]. Consequently, the development of tools or protocols to verify the exclusive use of organic fertilizer material in organic farming has become a priority to help maintain integrity of organic production [9].

Organic systems typically rely on organic sources of N, derived from animal or plant matter, such as by-products of fish and food processing, animal manures, biological N fixation by legumes, and other organic amendments like compost [10]. This contrasts with conventional systems, which utilize synthetic fertilizers to supply N in available inorganic forms. Organic N sources undergo microbial mineralization and nitrification to be converted into plant-available N forms such as ammonium ($NH_4^+$) and nitrate ($NO_3^-$) [11–14]. Proteins and their constituents, peptides and amino acids, are the main N source in organic fertilizers. Due to enzymatic reactions that discriminate against the heavier isotope during biochemical transformations, organic N sources often exhibit enrichment in $^{15}N$ ($\delta^{15}N > 0$) [15]. Conversely, conventional agriculture primarily depends on synthetically produced N fertilizers, which are manufactured with the Haber-Bosch process to convert atmospheric dinitrogen into ammonia. The $\delta^{15}N$ values of synthetic fertilizer N sources closely mirror that of atmospheric N ($\delta^{15}N = 0$).

Often described as "nature's ecological recorder", stable isotopes are widely used to provide insights into plant resource use, trophic food web interactions, and as biochemical and environmental probes to disentangle various temporal and spatial scales during plant and animal growth [16–19]. The stable isotope signature (e.g., $\delta^{13}C$ or $\delta^{15}N$), defined as the ratio of an element's stable isotope in relation to an international reference standard, can vary based on factors such as the element's initial concentration, the biochemical discrimination processes it undergoes, and environmental conditions [20–24]. This variability in the stable isotope signature is particularly noticeable among different crop production inputs, especially in fertilizer materials containing N, a critical plant macronutrient that significantly regulates plant productivity [25,26]. This makes it possible to use the $\delta^{15}N$ value of produce to indicate the specific N-based fertilizing material used in a cropping system.

Given the distinct differences in the N isotopic signature ($\delta^{15}N$) between synthetic and organic fertilizer N sources, this metric has been used to discriminate between organic and synthetic fertilizers [15] and, therefore, has potential to distinguish the sources of applied N fertilizer in crop from the different production systems [27–29]. Bueno *et al.* [30] demonstrated that the $\delta^{15}N$ value of organically grown tomatoes was higher than conventional production systems. A study investigating isotopic compositions of the soil inorganic N pool after the application of compost or fertilizer found that its $\delta^{15}N$ value ranged from −3 to + 2‰ in soils that received fertilizer, while compost-treated soils exhibited a value greater than + 8‰ [31]. This suggests that the $\delta^{15}N$ value of organically grown plants (i.e., those received compost) would be higher than that of conventionally grown plants (i.e., those treated with synthetic fertilizer), given that inorganic N is the most abundant form of N available for plant uptake. Although the $\delta^{15}N$ value of produce may serve as an indicator of organic or synthetic

fertilizer sources in crop production, broader investigations encompassing various production regions and crop species are necessary. Further considerations should include other potential indicators closely related to N cycling in cropping systems, such as carbon (C) content and its isotopic signature ($\delta^{13}$C). These investigations are necessary to evaluate the potential of these stable isotopic indicators as screening tools to verify the authenticity of organic labeled produce [32].

This study aims to evaluate the use of C and N contents and their stable isotope signatures as screening tools to discriminate between organic and conventional crops using a mixed model approach. We examined $\delta^{15}$N, $\delta^{13}$C, total N, total C, and C:N ratio of imported and locally produced fresh produce samples found across California markets to evaluate the authenticity of organic produce. The collection of a large number of samples is crucial for the development of robust predictive and explanatory models.

## Materials and methods

### Sample collection

Fresh samples of organically or conventionally grown produce were purchased from January through December of 2018 from food markets located across Fresno, Monterey, Salinas, and Sacramento Counties in California (n = 791). All samples were kept on ice while transported to the laboratory and stored in a freezer until preparation and analyses. Fresh produce geographical region of origin included the U.S., Mexico, Canada, Chile, China, Guatemala, Ecuador, Serbia, Peru, and Poland. Nearly 97% of the food samples analyzed were grown in Mexico and the U.S. (Fig S1A in S1 File). The remaining countries were grouped as "others" for the analysis. A similar number of organic and conventional fresh produce were collected (Fig S1B in S1 File, Table 1). The fresh produce samples represented 99 commercial food producers (companies). Seven major organically grown crops were selected for the study and consisted of berries (strawberry (*Fragaria × ananassa*), blueberry (*Vaccinium* sect. *Cyanococus*), blackberry (*Rubus* subg. *Rubus*), and raspberry (*Rubus idaeus*)) and vegetables (broccoli (*Brassica oleracea* var. *italica*), celery (*Apium graveolens*), and lettuce (*Lactuca sativa* var. *longifolia*)) (Fig S1C in S1 File).

### Elemental composition and stable isotope analysis

Samples were blended in a food processor, freeze-dried, and ground into powder with a mortar and pestle for elemental analysis of total C and N content and relative abundance of $\delta^{13}$C and $\delta^{15}$N using a combustion analyzer (Elementar Analysensysteme GmbH, Hanau, Germany) interfaced to an isotope ratio mass spectrometer (PDZ Europa 20-20 IRMS, Sercon Ltd., Cheshire, U.K.) at the Stable Isotope Facility, University of California, Davis.

Table 1. **Number of the organic and conventional fresh produce samples collected by crops.**

| Crop | Production system | |
|---|---|---|
| | Conventional | Organic |
| Blackberry | 45 | 36 |
| Blueberry | 100 | 93 |
| Broccoli | 65 | 56 |
| Celery | 32 | 70 |
| Lettuce | 34 | 64 |
| Raspberry | 33 | 34 |
| Strawberry | 60 | 69 |

Isotope abundances are expressed in δ notation as parts per mil (‰) and calculated as $\delta^{13}C = [((^{13}C/^{12}C)_{sample}/(^{13}C/^{12}C)_{standard}) - 1] \times 1000]$ and $\delta^{15}N = [((^{15}N/^{14}N)_{sample}/(^{15}N/^{14}N)_{standard} - 1) \times 1000]$.

## Statistical analysis

To evaluate the potential of using either elemental contents (e.g., C, N, C/N ratio) and or stable isotopes (e.g., $\delta^{13}C$, $\delta^{15}N$) for differentiating conventional and organic production, Holm corrected Welch t-tests and Tukey mean difference tests were applied to compare the mean values of these predictors between different crops and from different regions. Visualizations and data manipulation throughout use the "tidyverse" packages [33] in the R statistical computing language [34] (R Core Team, 2022).

A generalized linear mixed model (GLMM) with binary responses (logit link) of conventional vs. organic produce was used as the fundamental regression model. The classification model thresholds fitted probabilities from the regression model for prediction. This method was chosen for the ideal balance of classification flexibility, interpretability, and power. Predictor variables were selected by a combination of graphical examination, and automated model selection with Akaike Information Criteria, corrected (AICc) as the objective to minimize. AICc was chosen to discourage overfitting of the model and optimize for interpretability alongside performance. Model selection up to the second-order interaction across nine variables relating to elemental contents, stable isotopes, and source of crop were considered. GLMM fitting was achieved by adaptive Gaussian Quadrature for maximum likelihood, with the package "GLMMadaptive" [35]. The top 5 models were manually reviewed with a battery of graphical diagnostics (residual and influence plots), hypothesis testing (Osius-Rojek, Hosmer-Lemeshow, Stukel's goodness-of-fit), numerical stability (optimization parameters), and basic classification metrics (ROC curves, sensitivity, specificity, and accuracy).

The final chosen model is a function of C/N ratio, C percentage, $\delta^{15}N$, Crop type, and Company:

$$logit\left(E[Y]\right) = Intercept + Crop + CN + C\% + \delta^{15}N + Crop \times \delta^{15}N + Crop \times CN + \left(Company\right)$$

where E[Y] is the expected probability of a data point being observed as organic, and parentheses denote a random effect on the intercept. "Company" is modeled as a random effect to reflect the assumption that each producer company can have specific effects on the crop properties but belong to a more general population of companies that may not be represented in the data. This has advantages especially when some companies have a limited number of samples by "borrowing information" from companies with more data.

Diagnostics of the fitted model show several influential points around the extremes of the fitted distribution, likely due to the sampling structure of the dataset. There is poor fit around the logistic function in the 3rd and 7th deciles of fitted probability ranges, $\left(2.166e^{-9}, 0.01\right)$ and $\left(0.998, 1-1e^{-10}\right)$ respectively. The main effect coefficients of celery and blackberry appear the most sensitive to optimization parameters (~1 se), all other coefficients are sufficiently stabilized with 11 quadrature points. The observed tails of the company random effect are slightly larger than expected from the normal distribution, but not large enough for the normality assumption to be inappropriate (Shapiro-Wilk $p = 0.354$). Variable importance in the model is quantified by (hierarchical) term deletion (Rao) score test statistics in the fully fixed model, i.e., to evaluate the importance of C/N ratio, C/N ratio is removed from the model as well as any other higher order interactions involving C/N ratio.

The non-linearity of the link function and random effect of "Company" produce two sets of estimate targets, either coefficients conditional on the value of "Company" or with random effects marginalized out. These targets are commonly described as "subject-specific" (SS) models or "population-averaged" (PA) models in biostatistical literature [36]. We opt for the PA interpretation because the aggregate response of the entire population is our focus. Marginalized coefficients are estimated by Monte Carlo integration, using robust standard errors to protect against possible likelihood misspecification.

Overall quality of classification model is evaluated with areas under the receiver operating characteristic curve (AUROC). Searching for appropriate cutoff values for classification should be "Company" agnostic with the primary goal of < 1% false negative rate (FNR) of misclassification with "organic" as the positive case. FNR control is chosen to avoid penalizing honest organic farmers, and flag cases that are most likely conventionally grown but are labeled as organic. Prediction accuracy is used as a secondary characteristic for evaluating model usefulness. Separate effect profiles for each crop are created at the dataset average levels of C/N ratio, C percentage, and $\delta^{15}$N by crop, and 95% level confidence intervals for estimated probability are generated by normal approximation. For practicality of summarizing results, we consider single dimensional cutoffs with other variables held at crop specific averages but note that presence of interaction effects imply relative values of cutoffs may differ across different profiles of predictive variables. Classification criteria is developed for each crop, and different threshold values for the regression model are decided by taking the highest predicted probability that will produce a classification with FNR < 1% with 95% confidence. The confidence intervals around the FNR were generated by nonparametrically bootstrapping the dataset with 2000 replicates.

## Results

### Total C and N content, and C/N ratio

Carbon content varied among crops, with no differences among production systems or geographic origin (Table S1 in S1 File). Overall, carbon content ranged from 31.7 to 48.2%. Among the crops, celery contained the lowest average C (35.4%), and raspberry had the highest C content (43.8%). As indicated by the standard deviation, the highest C content variation was found in raspberry and blackberry (1.7% and 1.8%, respectively) (Fig 1b).

Nitrogen content was significantly different among crops and geographic regions ($p <$ 0.0001); however, there was no evidence of difference between production systems, except for celery for which higher N content was found in conventional systems than in organic systems (Fig 1a). More variability in mean N content was found among crops, ranging from 0.2 to 6.0%, and followed the order: blueberries (0.5%) < blackberries (1.1%) < strawberries (1.2%) < raspberries (1.3%) < celery (1.7%) < lettuce (3.4%) < broccoli (4.6%). Among geographic origin, N content was highest in the U.S. (2.27%), followed by Mexico (1.79%) and other countries (0.90%). Within geographic origin, only the U.S. showed significant differences in N content between conventional and organic systems ($p <$ 0.0001) (Table 2).

The C/N ratio was significantly different between crops and geographic origin ($p <$ 0.0001), with no differences between production systems (Table S1 in S1 File). Overall, vegetables had a lower C/N ratio compared to berries (Fig 1c): broccoli (10.4) < lettuce (12.1) < celery (23.2) < strawberries (36.1) < raspberries (37.2) < blackberries (40.5) < blueberries (81.8). Within geographic origin, the U.S. had significant differences in C/N ratio between organic and conventional systems in celery ($p <$ 0.0001).

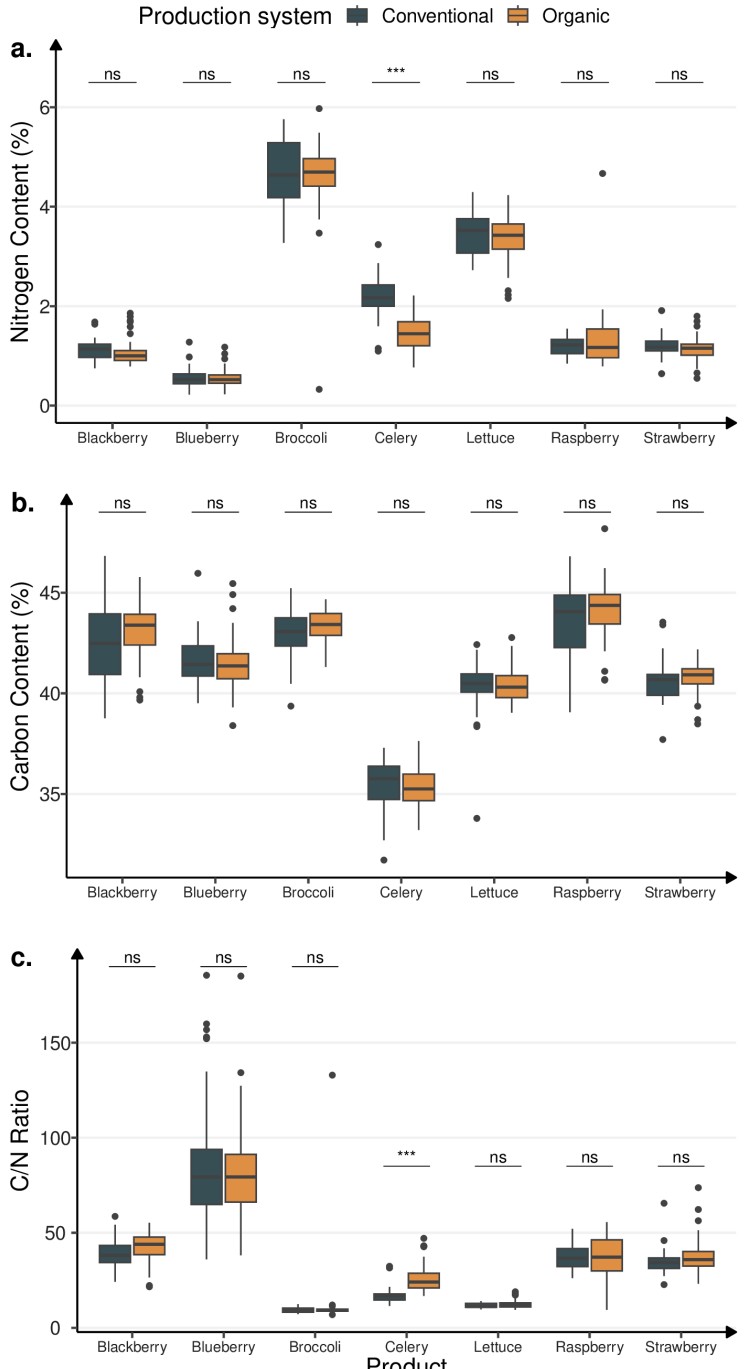

**Fig 1. Boxplots of nitrogen (a) and carbon (b) content (%), and C/N ratio (c) of conventional and organic crops.** Bars above boxplots denote a Welch two-sample t-test between conventional and organic crop, with adjusted $p < 0.001$ (***), $p < 0.01$ (**), $p < 0.05$ (*), and non-significant otherwise (ns). $p$-values are Holm corrected by each family defined by tests within the same subfigure.

## Natural δ¹³C abundance

Significant differences in δ¹³C value were found among geographic origin, production systems and crops ($p < 0.001$). Overall, δ¹³C values ranged from −30.95‰ to −21.28‰. Generally,

crops grown in Mexico exhibited higher natural $\delta^{13}$C abundance than that grown in the U.S. and the rest of the countries (Table S1 in S1 File). Overall, organic produce revealed a slightly, but significantly lower $\delta^{13}$C than conventional produce (Fig 2a, Table 2). Within a geographic origin, only Mexico had significant differences in $\delta^{13}$C values between organic and conventional crops ($p$ = 7.41e-07) (Table 2). Besides, the analyses of intrinsic differences among the crops showed that vegetables (broccoli, celery, and lettuce) had lower $\delta^{13}$C values than berries (blackberry, blueberries, raspberry, and strawberry) (Fig 2b, Table S1 in S1 File).

## Natural $\delta^{15}$N abundance

The $\delta^{15}$N values ranged from −4.2 to + 17.5‰, with an overall average and standard deviation of 4.04 ± 3.54‰. The $\delta^{15}$N values also varied among crops as: blueberries < broccoli < strawberry < raspberry < blackberry < lettuce < celery (Fig 2d). The geographical origin and

**Table 2. Isotopic composition of C ($\delta^{13}$C, ‰) and N ($\delta^{15}$N, ‰) by production system and geographic origin.** Entries are in the format "mean (sd) letter". Letter displays show Tukey range test significance for each isotope, among countries (columns). Different letters indicate the group difference of means is significantly different than 0 at the 0.05 level. The *p*-value column shows a Welch t-test between production systems within each country for each measured variable (rows).

| Isotope | Geographic Origin | Conventional | | Organic | | *p*-value |
|---|---|---|---|---|---|---|
| $\delta^{13}$C | USA | −26.75 (1.50) | a | −26.49 (1.48) | b | 0.113 |
| | Mexico | −25.31 (1.36) | b | −25.98 (1.19) | c | 7.41e-07 |
| | Other | −26.61 (1.49) | a | −27.34 (1.37) | a | 0.0246 |
| $\delta^{15}$N | USA | 1.18 (2.15) | a | 6.74 (3.12) | b | <2e-16 |
| | Mexico | 2.63 (2.16) | b | 6.45 (2.93) | b | <2e-16 |
| | Other | 1.28 (1.07) | a | 1.07 (1.96) | a | 0.531 |

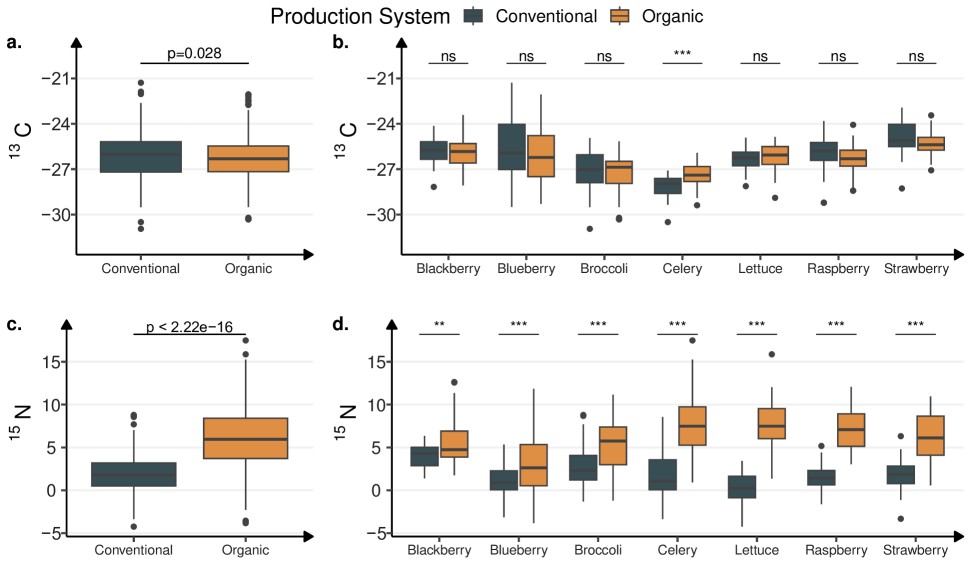

**Fig 2. Stable isotope ratio of $\delta^{13}$C (a and b) and $\delta^{15}$N (c and d) of organic and conventional crops.** Bars above box-plots denote a Welch two-sample t-test between conventional and organic crop, with $p < 0.001$ (***), $p < 0.01$ (**), $p < 0.05$ (*), and non-significant otherwise (ns). *p*-values are Holm corrected by each family defined by subfigure.

production system significantly affected the $\delta^{15}$N values of the crops ($p < 0.001$). The crops grown in the U.S. and Mexico generally exhibited significantly higher average $\delta^{15}$N, with 3.9 ± 3.9‰ and 4.8 ± 3.2‰, respectively, than crops grown in other countries (Canada, Chile, China, Guatemala, Ecuador, Serbia, Peru, and Poland) with 1.2 ± 1.6‰ (Tables S1 and S2 in S1 File).

A strong separation in $\delta^{15}$N values was observed between conventional and organic systems for most crops (Fig 2c). Regardless of the differences in $\delta^{15}$N among the geographical origins, there was a three-fold increase in mean $\delta^{15}$N values of organic crops compared to conventionally produced crops (Fig 2c). Average $\delta^{15}$N value was 1.8 ± 2.2‰ for conventional and 5.9 ± 3.4‰ for organic crops. Celery, lettuce, raspberry, and strawberry showed the largest difference in $\delta^{15}$N values between conventional and organic systems.

## Predicting crop production system

In the fully fixed version of our final model, the ranking of highest variable importance (respecting the hierarchical principle by removing dependent higher order terms) is: company (score $\chi^2_{106} = 294.64$, $p < 1.0e^{-16}$), $\delta^{15}$N (score $\chi^2_7 = 168.45$, $p < 1.0e^{-16}$), crop (score $\chi^2_{18} = 61.28$, $p = 1.3e^{-06}$), C/N ratio (score $\chi^2_7 = 35.88$, $p = 7.7e^{-06}$), C %(score $\chi^2_1 = 8.91$, $p = 2.8e^{-03}$) (Table S2 in S1 File). The presence of crop × C/N ratio and crop × $\delta^{15}$N interactions imply each crop has a different relationship between C/N Ratio, $\delta^{15}$N and organic classification. For $\delta^{15}$N, the strongest slopes (log-odds scale) are for raspberry (slope = 3.36, $p = 6.95e^{-04}$), strawberry (slope = 0.80, $p = 9.66e^{-05}$), lettuce (slope = 0.35, $p = 0.242$) and celery (slope = 0.30, $p = 3.89e^{-05}$). For C/N Ratio, the strongest slopes (log-odds scale) are for raspberry (slope = −0.30, $p = 1.87e^{-04}$), broccoli (slope = 0.28, $p = 0.16$), strawberry (slope = 0.08, $p = .068$). C percentage across all crops has an estimated slope (log-odds scale) of 0.09 ($p = 0.11$) (Table S3 in S1 File).

In classification, the SS model of fitted probabilities give an AUROC of 0.983, while the PA model has an AUROC of 0.863, showing the drop in accuracy with a company agnostic classification mode (Fig 3). Since each crop's characterization is different for classification (Fig S2 in S1 File), controlling the FNR < 1%, we get different probability thresholds and performance metrics for each crop (Table S4 in S1 File). Holding C % and C/N ratio at their crop specific

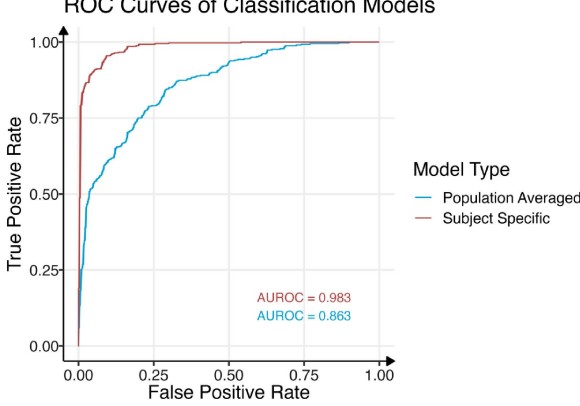

**Fig 3. ROC curves based on predicted probabilities of organic classification from GLMM model across dataset.** Population averaged and subject specific models for "company" are both shown. The drop in AUROC between the models show the loss of company information in classification. The tradeoff between true positive rate (TPR) and false positive rate (FPR) is relatively symmetric in the tails. We focus our attention on regions of high TPR for developing crop specific cutoffs for covariates.

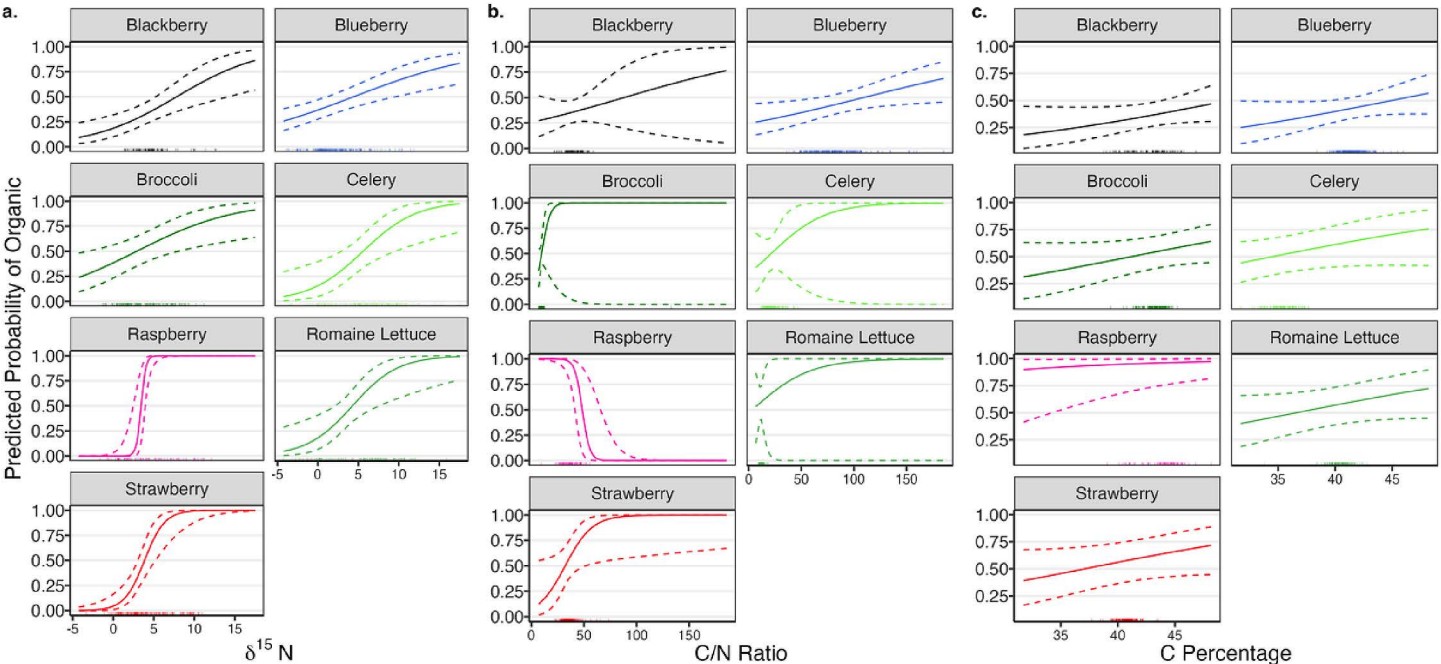

**Fig 4. Effect profile plots of (a) δ15N, (b) C/N Ratio, (c) C percentage with crop specific sample averages for other covariates.** Fitted probabilities are population-averaged across company/regions. 95% confidence intervals (normal approximation) are shown for fitted probability of organic classification in dotted lines. Marginal sample distributions of covariates for each crop are shown as ticks across the bottom axis.

sample means, the cutoff values were $\delta^{15}N < 2.17$ ‰ for raspberries and $\delta^{15}N < 3.22$ ‰ for strawberry. Similarly, holding C % and $\delta^{15}N$ at crop specific averages, raspberries with C/N ratio > 69.2 should be flagged as suspiciously non-organic. Other crops and covariates do not reach the < 1% FNR threshold standard for univariate cutoffs for the average crop (Fig 4).

## Discussion

### The impact of production systems on crop elemental compositions

We found consistently greater $\delta^{15}N$ values in organic (5.98 ± 3.38 ‰) than in conventional (1.84 ± 2.19 ‰) systems (Table 2). Classification analysis indicated that $\delta^{15}N$ natural abundance was the biological variable with the highest predictive capability for distinguishing between organic and conventional food production systems. This consistent difference aligns with previous studies [27,30], which report higher $\delta^{15}N$ values in organic crops compared to conventional crops. The elevated $\delta^{15}N$ values in organic farming systems are primarily attributed to the exclusive use of organic fertilizers and amendments such as manure or compost, which inherently possess higher $\delta^{15}N$ signatures [15]. In contrast, conventional farming systems predominantly rely extensively on synthetic fertilizers, which typically have lower $\delta^{15}N$ values [15]. While the occasional use of organic amendments such as manure or compost in conventional systems may lead to higher $\delta^{15}N$ values compared to exclusive synthetic fertilizer use, the $\delta^{15}N$ values of certified organic produce are generally expected to remain above those of conventional counterparts due to the prohibition of synthetic fertilizers in organic farming. Contamination from adjacent conventional fields may also contribute to $\delta^{15}N$ variability in organic produce, potentially narrowing the differences between organic and conventional systems. However, such contamination is unlikely to cause $\delta^{15}N$ signatures

of certified organic produce to fall below those commonly observed in conventional systems. Therefore, although the use of a single $\delta^{15}N$ threshold may seem arbitrary and does not fully capture variations in climate regions, soil conditions, land management practices, or organic fertilizer sources, $\delta^{15}N$ remains a promising preliminary assessment tool. Specifically, it can help identify food products with exceptionally low $\delta^{15}N$ values that warrant further investigation to verify the authenticity of organic labels.

Overall, differences in $\delta^{15}N$ between production systems were more prominent in broccoli, celery, lettuce, raspberries, and strawberries (Fig 2). Blackberries showed little differences in $\delta^{15}N$ abundance between production systems, being relatively high in both types of farming systems, which is possibly due to the heavy use of natural N sources (bloodmeal, feather meal, composted manure) in both organic and conventional blackberries [37]. A study in a selected group of vegetables (potato, leek, cauliflower, tomato) in Slovenia found that the N stable isotope ratio was a successful technique to identify production practices [38]. Vegetables like garlic, onion, parsley, sweet pepper, and carrots showed $\delta^{15}N$ values overlapped between conventional and organic production practices. In the Slovenian study, the differences in $\delta^{15}N$ values found between organically and conventionally grown products ranged from 3 to 6‰; nevertheless, it is essential to mention that the analyzed sample size was relatively small (n = 107, approx. 3–6 samples per vegetable). A larger sample size, as conducted in the current study (n = 791), could reduce the dispersion in $\delta^{15}N$ values, which are known to vary widely by species, soil management (e.g., N input, crop rotation), and environmental conditions [39]. For example, in an extensive study analyzing over 100 soils from 20 U.S. states, the $\delta^{15}N$ of soil was found to vary from 5 to 12‰, influenced by factors such as climate, depth, soil pH, and land use [40].

The C and N content of berries and vegetables were not significantly different between the two production systems (organic and conventional), except conventionally grown celery showed a higher average N content than organic (2.17% vs. 1.45%). Similar findings were reported in celery, in which the high N content was found in conventional grown celery after supplying a high amount of inorganic N, mainly as ammonium ($NH_4^+$) [41]. After further analyzing the N forms in the previous study, it was found that higher $NH_4^+$ content occurred in the conventional grown celery than the organic grown celery, while nitrate ($NO_3^-$) content was not significantly different between these two production systems.

## Predicting cropping systems and perspectives

The elemental composition analysis of food has been demonstrated in several studies to differentiate between crop production systems from different geographical origins [9,28,30]. However, to the best of our knowledge, no study has attempted to determine a threshold for specific crops, possibly due to the limitations in the sample size used in these studies. In the current study, an extensive crop sampling (n = 791) was conducted to select variables that could help to verify the authenticity of organic production of organic berries and vegetables.

The covariates used in the final fitted model were evaluated by associative strength and do not imply a causal relationship. Nor does it exclude other variables from potentially having strong relationship with organic and conventional classification. "Company" emerges as the strongest predictor in these models, surpassing any biological measurement. This is understandable considering that some companies exclusively produce organic crops, while others focus solely on conventional crops. Moreover, some companies produce specific subsets of crops, and estimating the company effects for crops they do not produce requires imputation. Knowing the crop company would thus significantly bias the prediction one way or another. Our cross-sectional summaries displayed relatively good coverage of both crops and organic systems. However, developing screening criteria should be company agnostic; thus, these

considerations point to modeling the company as a more general population. The variance structure was not deeply explored in this application and would require more structured sampling and larger datasets for numerical feasibility. We did observe slightly larger variability in the actual dataset than was assumed during modeling. A deeper exploration of the source of this variability could yield more accurate models.

Although model misfit was observed in regions of extreme probability, the increased variability in these regions probably won't affect the classification problem application. The increased variability in these extremes is largely due to data points from sparsely sampled companies or extreme values of biological measurements. All other regions of the model fit well and have been shown to reliably differentiate between organic and conventional crops. These biases and the dependence on dataset sampling should be considered when interpreting model results.

From this dataset, we have found compelling evidence for $\delta^{15}N$ to be used as screening criteria for classifying organic and conventional systems in raspberries and strawberries (Fig 4). The cutoffs developed are in tail regions of our data as expected, but this makes precision of lower FNR control harder to scrutinize. Increased systematic sampling, i.e., examples of truly, fraudulently grown crops, would further ensure that the classification criteria suggested is robust to any unmeasured confounders. There is insufficient evidence for many of the other crops to achieve low levels of false negatives due to their weaker distinguishing features. We note that the relationship of lettuce with $\delta^{15}N$ appears to be robust and would benefit from a further investigation with an increased sample size. C/N ratio also has a strong association with broccoli, raspberries, and strawberries, indicating that these covariates have potential to serve as a useful criterion for classification. Interestingly, raspberries display a trend where higher C/N ratios reduce the chance of being classified as organic – a pattern only evident when conditioned on C % and $\delta^{15}N$ and absent in marginal distributions, meriting further investigation.

Our findings suggest that by implementing crop elemental composition analysis – particularly N stable isotopic analysis – it is feasible to predict the food production system at the market level. The classification tool we employed can pinpoint produce potentially fertilized with synthetic N sources, guiding further authenticity investigations. Although distinct differences in $\delta^{15}N$ among produces were observed and preliminary crop-specific thresholds set, the N stable isotope ratio's variance associated with soil organic matter and climate conditions [22,42] might constrain its broader application. Future studies should focus on structured sampling with a larger dataset to investigate relationships among soil properties, climatic conditions, land management, and crop elemental compositions to better develop and apply such screening tools.

## Conclusions

Our study showed that the total C and N content, as well as the C/N ratio, showed little variation between production systems and geographical origins for most crops. It's particularly significant to note that $\delta^{15}N$, with observed values averaging 1.8 ± 2.2‰ for conventional crops and 6.0 ± 3.4‰ for organic, emerged as the most consequential predictor in distinguishing production systems. Indeed, organic produce consistently exhibited higher $\delta^{15}N$ values. Models were developed for seven crops with isotope data for which strawberries, raspberries, lettuce, and celery were identified as the most promising candidates for classification of production methods based on these models. A $\delta^{15}N$ threshold is suggested to differentiate conventional from organic grown raspberries ($\delta^{15}N$ < 2.17‰) and strawberries ($\delta^{15}N$ < 3.22‰), for an estimated false negative rate of 1%. While further investigations are necessary

to solidify the $\delta^{15}N$ thresholds between organic and conventional production methods, our extensive dataset of 791 samples provides invaluable insights. It not only sheds light on the characteristics of different agricultural production systems but also lays the groundwork for the enhancement of organic verification processes in the future.

## Supporting information

**S1 File. Supplementary for "Stable isotopes as a predictor for organic or conventional classification of berries and vegetables".**
(PDF)

## Acknowledgment

We thank Eshan R. Toosi and Martin Burger for their comments on earlier versions of this manuscript.

## Author contributions

**Conceptualization:** Xia Zhu-Barker.

**Data curation:** Xia Zhu-Barker, Michael Liou, Diana Zapata.

**Formal analysis:** Xia Zhu-Barker, Michael Liou, Diana Zapata.

**Funding acquisition:** Xia Zhu-Barker, William R. Horwath.

**Investigation:** Xia Zhu-Barker.

**Methodology:** Xia Zhu-Barker, Michael Liou, Diana Zapata.

**Project administration:** Xia Zhu-Barker.

**Resources:** Xia Zhu-Barker, William R. Horwath.

**Software:** Michael Liou, Jingyi Huang.

**Validation:** Michael Liou, Diana Zapata, Xia Zhu-Barker, Jingyi Huang, William R. Horwath.

**Visualization:** Michael Liou, Diana Zapata, Jingyi Huang.

**Writing – original draft:** Xia Zhu-Barker.

**Writing – review & editing:** Michael Liou, Diana Zapata, Jingyi Huang, William R. Horwath.

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
