## [Decision Letter · Decision Letter 0]

20 Sep 2024

PONE-D-24-32483Stable Isotopes as a Predictor for Organic or Conventional Classification of Berries and VegetablesPLOS ONE

Dear Dr. Zhu-Barker,

Thank you for submitting your manuscript to PLOS ONE. After careful consideration, we feel that it has merit but does not fully meet PLOS ONE’s publication criteria as it currently stands. Therefore, we invite you to submit a revised version of the manuscript that addresses the points raised during the review process.

We look forward to receiving your revised manuscript.

Kind regards,

Taimoor Hassan Farooq

Academic Editor

PLOS ONE

“The study was supported by the J.G. Boswell Endowed Chair in Soil Science, the Wisconsin Dairy Innovation Hub, and the Office of the Vice Chancellor for Research and Graduate Education at the University of Wisconsin-Madison with funding from the Wisconsin Alumni Research Foundation. We thank Eshan R. Toosi and Martin Burger for their comments on earlier versions of this manuscript”

5. We noted in your submission details that a portion of your manuscript may have been presented or published elsewhere. [The manuscript has been published as a preprint at Research Square: https://doi.org/10.21203/rs.3.rs-3690228/v2.] Please clarify whether this publication was peer-reviewed and formally published. If this work was previously peer-reviewed and published, in the cover letter please provide the reason that this work does not constitute dual publication and should be included in the current manuscript.

6. Please note that your Data Availability Statement is currently missing the repository name and/or the DOI/accession number of each dataset OR a direct link to access each database. If your manuscript is accepted for publication, you will be asked to provide these details on a very short timeline. We therefore suggest that you provide this information now, though we will not hold up the peer review process if you are unable.

Reviewers' comments:

Reviewer's Responses to Questions

**Comments to the Author**

1. Is the manuscript technically sound, and do the data support the conclusions?

Reviewer #1: Yes

Reviewer #2: Yes

2. Has the statistical analysis been performed appropriately and rigorously? 

Reviewer #1: Yes

Reviewer #2: Yes

3. Have the authors made all data underlying the findings in their manuscript fully available?

Reviewer #1: Yes

Reviewer #2: Yes

4. Is the manuscript presented in an intelligible fashion and written in standard English?

Reviewer #1: Yes

Reviewer #2: Yes

5. Review Comments to the Author

Reviewer #1: The manuscript is well-designed and well-prepared. only some comments in the attached reviewed manuscript.

Also the manuscript could be very benefits from an English editing to improve the flow of the text and rephrasing some sentences

Reviewer #2: Dear Authors,

Thank you for the opportunity to contribute to your work as a reviewer. This study aims to introduce a methodology for the differentiation of organic and non-organic produces with the quantification of N stable isotopes. The experiment is well designed and the results are visualized in an understandable way. However, clarifications needed when interpreting the results, and there is space for further explanations.

General comments

1. It is a common misbelief, that organic farming is organic just because of the fact, that the farmers do not use chemicals. Instead, it is organic, because the certification body says, based on the regulatory background, that it is organic. Therefore, it is not clear, what the sentence in line 19 is about – please revise. Also, it would be useful to add a clarifying sentence about the USDA/EU regulatory background on organics.

2. Regarding the collection of samples, it is unclear for me, when did the authors purchased the vegetables and berries. It is reported, that it took a whole year, but no information is available about the exact time or the production system those were grown. All of the samples were grown on genuine soil? Were those from uncontrolled/controlled environment facilities, hydroponics, aeroponics, etc? Were those fresh or stored products? And, most importantly, within species, were those purchased in more or less the same time? Obviously, when comparing a CEA lettuce harvested in January and an open-field one harvested in October does not seem fair. At the same time, I would like to emphasize, that I do not doubt the results due to the high number of samples, but I find it important to add more general information about the origin of the produces (not detailed information one by one!) and clarifying the possible issues arising from that.

3. The Authors mention, that this method can be used to demonstrate unauthorized use of synthetic fertilizers in OF. What is the amount they think it is possible to detect the difference between OF and non-OF products? I am asking it as the results showed, that the most robust difference was found in the case of lettuce, which is known to be produced with the use of high doses of synthetic fertilizer in conventional farming, therefore the difference can be high comparing with OF.

4. I am unsure whether the differentiation is about OF versus non-OF, as the farmer should not be a certified organic farmer for the use of manure or compost. Again, organic is about the certification provided by the authorization body. The authors might want to deal briefly with this question in the ms. Additionally, as the breakdown products of any chemicals used in the field remain in the soil and might travel with soil water, OF produces can also be polluted with these materials, without breaking the regulation, especially in areas where heavy agrochemization is an issue.

Detailed comments

line 13: significantly?

line 47: What about the EU? Or add: For example, (…)

line 49: It is not the certification process, that leads to higher production costs, but the exclusion of cheap agrochemicals. Please revise.

line 54: Revise the sentence starting with “This prohibition …

line 84-85: Revise.

line 120: Please use complete scientific names, or at least add sp.

line 263: Was that romaine lettuce, or any type of lettuce? Clarify throughout the whole ms.

line 269: ’Company’ within inverted commas?

line 295: There are several factors besides climate and soil conditions, that counts, please add more.

line 307: Slovenian?

6. PLOS authors have the option to publish the peer review history of their article (what does this mean? ). If published, this will include your full peer review and any attached files.

**Do you want your identity to be public for this peer review?** For information about this choice, including consent withdrawal, please see our Privacy Policy .

Reviewer #1: No

Reviewer #2: No

---

## [Author Response · Author response to Decision Letter 1]

15 Nov 2024

We have uploaded a revised manuscript with track changes along with a manuscript with no-change tracking that takes into consideration the reviewer comments which we found to be constructive and helpful. We have also uploaded a response letter that detailed our point-by-point responses to the reviewers’ comments with references to line numbers in the revised version. Let me know if further information or modifications are needed.

We have also detailed our responses to Journal's requirements in the cover letter

---

## [Decision Letter · Decision Letter 1]

12 Jan 2025

Stable Isotopes as a Predictor for Organic or Conventional Classification of Berries and Vegetables

PONE-D-24-32483R1

Dear Dr. Zhu-Barker,

We’re pleased to inform you that your manuscript has been judged scientifically suitable for publication and will be formally accepted for publication once it meets all outstanding technical requirements.

Kind regards,

Taimoor Hassan Farooq

Academic Editor

PLOS ONE

Additional Editor Comments (optional):

Dear authors,

I am pleased to inform you that your manuscript has been accepted for publication in PLOS One. After careful consideration and review, we are confident that your work will be a valuable contribution to the field.

Please ensure that all necessary revisions and final formatting are completed in accordance with the journal's guidelines.

Reviewers' comments:

Reviewer's Responses to Questions

**Comments to the Author**

1. If the authors have adequately addressed your comments raised in a previous round of review and you feel that this manuscript is now acceptable for publication, you may indicate that here to bypass the “Comments to the Author” section, enter your conflict of interest statement in the “Confidential to Editor” section, and submit your "Accept" recommendation.

Reviewer #1: All comments have been addressed

Reviewer #2: All comments have been addressed

2. Is the manuscript technically sound, and do the data support the conclusions?

Reviewer #1: Yes

Reviewer #2: Yes

3. Has the statistical analysis been performed appropriately and rigorously? 

Reviewer #1: Yes

Reviewer #2: Yes

4. Have the authors made all data underlying the findings in their manuscript fully available?

Reviewer #1: Yes

Reviewer #2: Yes

5. Is the manuscript presented in an intelligible fashion and written in standard English?

Reviewer #1: Yes

Reviewer #2: Yes

6. Review Comments to the Author

Reviewer #1: The author responded to all reviewer's comments. The manuscript can be accepted now if there no negative comments from other reviewers

Reviewer #2: Dear Authors, thank you for considering my suggestions for the improvement of the manuscript, I can accept all of your responses.

7. PLOS authors have the option to publish the peer review history of their article (what does this mean? ). If published, this will include your full peer review and any attached files.

**Do you want your identity to be public for this peer review?** For information about this choice, including consent withdrawal, please see our Privacy Policy .

Reviewer #1: No

Reviewer #2: No

---

## [Editor Report · Acceptance letter]

PONE-D-24-32483R1

PLOS ONE

Dear Dr. Zhu-Barker,

I'm pleased to inform you that your manuscript has been deemed suitable for publication in PLOS ONE. Congratulations! Your manuscript is now being handed over to our production team.

Kind regards,

on behalf of

Taimoor Hassan Farooq

Academic Editor

PLOS ONE